

# Photochemical impacts of haze pollution in an urban environment

Michael Hollaway[1], Oliver Wild[1], Ting Yang[2], Yele Sun[2,3], Weiqi Xu[2,3], Conghui Xie[2,3], Lisa Whalley[4,5], Eloise Slater[4], Dwayne Heard[4,5], and Dantong Liu[6,7]

[1]Lancaster Environment Centre, Lancaster University, Bailrigg, Lancaster, UK
[2]State Key Laboratory of Atmospheric Boundary Layer Physics and Atmospheric Chemistry, Institute of Atmospheric Physics, Chinese Academy of Sciences, Beijing 100029, China
[3]University of Chinese Academy of Sciences, Beijing 100049, China
[4]School of Chemistry, University of Leeds, Leeds, UK
[5]National Centre for Atmospheric Science, University of Leeds, Leeds UK
[6]Department of Atmospheric Sciences, School of Earth Sciences, Zhejiang University, Hangzhou, Zhejiang, China
[7]Centre for Atmospheric Sciences, School of Earth and Environmental Sciences, University of Manchester, Manchester, UK

**Correspondence:** Michael Hollaway (m.hollaway@lancaster.ac.uk)

**Abstract.** Rapid economic growth in China over the past 30 years has resulted in significant increases in the concentrations of small particulates ($PM_{2.5}$) over the city of Beijing. In addition to health problems, high aerosol loading can impact visibility and thus reduce photolysis rates over the city leading to potential implications for photochemistry. Photolysis rates are highly sensitive not only to the vertical distribution of aerosols but also to their composition as this can impact how the incoming solar
radiation is scattered or absorbed. This study, for the first time, uses aerosol composition measurements and lidar optical depth to drive the Fast-JX photolysis scheme and quantify the photochemical impacts of different aerosol species during the Air Pollution and Human Health (APHH) measurement campaigns in Beijing in November–December 2016 and May–June 2017. This work demonstrates that severe haze pollution events ($PM_{2.5} > 75\,\mu gm^{-3}$) occur during both winter and summer leading to reductions in $O_3$ photolysis rates of 27.4–34.0 % (greatest in winter) and reductions in $NO_2$ photolysis of 40.4–66.2 %
(greatest in summer) at the surface. It also shows that in spite of much lower $PM_{2.5}$ concentrations in the summer months, the absolute changes in photolysis rates are larger for both $O_3$ and $NO_2$. In the winter absorbing species such as black carbon dominate the photolysis response to aerosols leading to mean reductions in $J[O^1D]$ and $J[NO_2]$ in the lowest 1 km of 23.8 % and 23.1 % respectively. In contrast in the summer, scattering aerosol such as organic matter dominate the response leading to mean decreases of 2.0–3.0 % at the surface and increases of 8.4–10.1 % at higher altitudes (3–4 km). During these haze
events in both campaigns, the influence of aerosol on photolysis rates dominates over that from clouds. These large impacts on photochemistry can have important implications for concentrations of important atmospheric oxidants such as the hydroxyl radical.

## 1 Introduction

As a result of rapid economic growth and industrialisation over the past 30 years, air pollution has become a major problem
in China (Chan and Yao, 2008; Zhang et al., 2015), with an increase in the number of haze episodes emerging as a particular issue. During such haze events, concentrations of small aerosol particles ($PM_{2.5}$: particles with an aerodynamic diameter





of less than 2.5 $\mu$m) can climb to very high levels (Han et al., 2015; Wang et al., 2018) leading to significant reductions in visibility, health problems and potential feedbacks on atmospheric chemistry and dynamics (Cheng et al., 2011; Han et al., 2015; Lelieveld et al., 2015; Xing et al., 2017).

High aerosol loadings not only impact dynamics through their regulation of the atmospheric radiation budget (Kaiser and Qian, 2002; Hu et al., 2017), but can also have significant impacts on atmospheric oxidation capacity through reductions in photolysis rates (Liao et al., 1999; Lou et al., 2014; Tang et al., 2003; Li et al., 2011; Xing et al., 2017). Photolysis plays a very important role in initiating atmospheric photochemistry. Of particular importance are the photolysis of $NO_2$ in the troposphere ($J[NO_2]$), which is key to chemical generation of ozone ($O_3$) and the photolysis of $O_3$ itself ($J[O^1D]$) to produce electronically excited oxygen ($O^1D$). $O^1D$ may subsequently react with water vapour and is the main source of the hydroxyl radical (OH) in the atmosphere globally. In addition, in highly polluted urban environments such as Beijing, photolysis of nitrous acid (HONO) is also a major source of OH. OH is highly reactive and serves as the primary oxidation sink of many atmospheric species whilst also playing a key role in initiating catalytic cycles that result in poor air quality (e.g. $O_3$ formation).

Both the vertical distribution and composition of aerosols can impact how incoming solar radiation is absorbed or scattered throughout the atmospheric column which in turn can significantly affect photolysis rates (Tang et al., 2003; Li et al., 2011). For example, Li et al. (2011) estimated that $J[O^1D]$ rates over Eastern China were reduced by 53 %, 37 % and 21 % in the lower, middle and upper troposphere during high summer aerosol loadings in 2006. This resulted in corresponding OH concentration reductions of 51 %, 40 % and 24 % respectively. Therefore, vertical characterisation of aerosols is critical to fully understanding the impact of severe haze on atmospheric photochemical processes. The key processes behind the formation and composition of severe haze events over China at ground sites (including in Beijing) have been studied extensively (Han et al., 2015; Huang et al., 2014; Ji et al., 2014; Zhao et al., 2013; Sun et al., 2013), with some studies also using aircraft and tethered balloon measurements to investigate the vertical profiles of aerosol and gaseous pollutants (Chen et al., 2009; Zhang et al., 2009; Ran et al., 2016; Li et al., 2015). Aircraft typically only permit measurements above 300 m altitude and fail to capture the full evolution of aerosol profiles and tethered balloons are normally operated in rural settings that are not representative of urban environments. Therefore, greater understanding of the vertical distribution of aerosol loadings is still required.

Recent studies have better captured the evolution of black carbon (BC) profiles and their associated optical impacts during haze events over Beijing (Wang et al., 2018), but these are still limited to the lower boundary layer (< 260 m altitude). Lidar instruments are useful in providing vertical profiles of total aerosol extinction and have been applied over Beijing previously (Yang et al., 2010, 2017) to measure up to a height of 6 km. However, only the total extinction can be retrieved and there is no information on the contribution from different aerosol species (e.g. strong absorbers such as BC or scattering aerosol such as ammonium sulphate, $(NH_4)_2SO_4$, which is required to understand how the incoming solar radiation is scattered and absorbed). As a result, atmospheric models are often used to simulate aerosol distributions and composition throughout the atmospheric column and their impacts on photolysis rates (Tang et al., 2003; Li et al., 2011). However, such model studies often are often poorly constrained by observations (particularly in the upper atmosphere) and can fail to accurately predict peaks in aerosol concentrations during severe haze events (Wang et al., 2014b, a, c; Zheng et al., 2015).



In this study, aerosol composition and lidar extinction measurements from two intensive field campaigns conducted in Beijing in Winter 2016 and Summer 2017 are used to derive chemically apportioned vertical profiles of aerosol extinction. These profiles are used, for the first time, to critically test an offline photolysis scheme against measurements of photolysis rates under observed aerosol loadings and to quantify the contribution of different aerosol components to changes in photolysis rates. This enables a better understanding of the impacts of severe haze episodes on pollutant photochemistry in contrasting seasons and provides insight into the photochemical implications of potential future pollution reduction strategies.

## 2 Materials and Methods

### 2.1 Measurement campaigns and sampling site

Two measurement campaigns were conducted at the tower site of the Institute of Atmospheric Physics, Chinese Academy of Sciences (IAP-CAS) as part of the joint UK-China Air Pollution and Human Health (APHH) programme addressing the sources, processing and impacts of air pollution in Beijing (Shi et al., 2018). The site is located in urban surroundings (39.6° N, 116.2° E) between the third and fourth ring roads in the North of Beijing, 40 m from the nearest road and around 400 m from the Jingzang Highway (Han et al., 2015). The first campaign was conducted from 5[th] November to 10[th] December 2016 and the second campaign was conducted from 15[th] May to 22[nd] June 2017 in order to monitor haze episodes during different seasons (Shi et al., 2018).

### 2.2 Instrumentation

Non-refractory aerosol species (including sulfate, nitrate, chloride, ammonium and organics) and black carbon (BC) were measured using an Aerodyne High-Resolution Time-of-Flight Aerosol Mass Spectrometer (HR-ToF-AMS; DeCarlo et al., 2006) and a 7-wavelength Aethalometer (model AE33; Magee Scientific Corp., Drinovec et al., 2015) respectively, which provided aerosol composition data at 5 minute resolution. Total aerosol extinction and aerosol absorption were measured simultaneously at 870 nm using a Photoacoustic Extinctiometer (PAX; Droplet Measurement Technologies, Boulder, CO, USA), with data reported for dry aerosol (approximately 40% relative humidity) at 5 minute resolution. The PAX, HR-ToF-AMS and Aethalometer were all located on the roof of a three floor laboratory building at the tower site.

Vertical profiles of aerosol extinction were obtained using a dual-wavelength (1064, 532 nm) depolarisation lidar which was located at a height of 28 m on the roof of a building near the aerosol monitoring equipment. The lidar provided extinction coefficients attributed to anthropogenic aerosols at 30 m vertical resolution up to an altitude of 6 km and 15 minute temporal resolution for both campaign periods. Further details of the lidar instrument and calibration procedures can be found in Yang et al. (2010, 2017) and Sugimoto et al. (2002).

The spectrally resolved ($\approx$ 1 nm) actinic flux was measured during both campaign periods at a temporal frequency of 1 minute using a spectrometer (Ocean Optics QE65000) which was fibre-coupled to a 2pi quartz receiver optic (Meteorologie Consult GmbH). J[O$^1$D] and J[NO$_2$] rates were then calculated using literature values of wavelength dependent photodissoci-



ation quantum yields and absorption cross-sections (Whalley et al., 2018). The instrument provided measurements representative of a height of 3.5 m above ground level.

Finally, a single particle soot photometer (SP2) instrument was used to measure the physical properties of individual black carbon (BC) particles during each campaign period. Described in detail by Liu et al. (2010, 2014), the SP2 employs a laser at
1064 nm to detect the optical properties of BC particles including core diameter and coating thickness for each single particle. The core size and coating information for BC covering both the winter and summer campaign periods is presented in detail in Liu et al. (2018a).

## 2.3 Chemical apportionment of aerosol extinction

To estimate the vertical extinction profile for each species, it is necessary to chemically apportion the total aerosol extinction
measured with the lidar in the 532 nm channel. The co-located PAX, HR-ToF-AMS and Aethelometer data are used here to develop an empirical relationship between aerosol composition and optical properties.

Based on the assumption that aerosol particles are externally mixed and that the extinction of individual species are independent of one another, the contribution of the non-refractory aerosol species to the total extinction coefficient ($b_{ext}$) is estimated at 870 nm. A differential evolution optimisation algorithm is utilised on the scattering coefficient ($b_{sct}$) and the respective con-
centrations of ammonium sulfate $(NH_4)_2SO_4$, ammonium nitrate $NH_4NO_3$, ammonium chloride $NH_4Cl$ and organic aerosol (OA). The concentration of the inorganic aerosol components was calculated from the measurements of sulfate, nitrate and chloride assuming that these ions are neutralised by ammonium.

The scattering coefficient $b_{sct}$ is assumed to be the difference between $b_{ext}$ and the absorption coefficient ($b_{abs}$) measured by the PAX instrument. As the algorithm is applied to lidar measurements in ambient conditions, the effects of aerosol hygroscopic
growth needs to be accounted for through use of the humidity dependent growth factor $f_{RH}$. This work uses the $f_{RH}$ factor from the Interagency Monitoring of Protected Visual Environments (IMPROVE) algorithm (Pitchford et al., 2007) which has previously been used to chemically apportion PM extinction over China (Shen et al., 2014). As in the IMPROVE algorithm, the effects of hygroscopic growth are only allowed to impact the inorganic ions. The differential algorithm is calibrated using an optimisation approach to minimize the mean absolute error between the observed scatter ($b_{sct}$) and the value estimated with
the differential evolution algorithm. This allows the mass scattering efficiency (MSE) for each species to be estimated.

For the contribution of BC to overall extinction, it is assumed that all absorption from the PAX measurements can be attributed to BC, following the approach of Han et al. (2015). A simple linear regression model is fitted between the absorption coefficient ($b_{abs}$) and the measured BC mass concentration. The slope of this regression yields the mass absorption efficiency (MAE) for BC. Combining the calculated MSE and MAE values for each PM constituent gives the following empirical rela-
tionships for the winter (Eqn 1) and summer (Eqn 2) campaigns.

$$
\begin{aligned}
b_{sct_{win}} = {} & 4.0 * f_{RH} * [(NH_4)_2SO_4] + 1.1 * f_{RH} * [(NH_4)NO_3] + \\
& 0.3 * f_{RH} * [(NH_4)Cl] + 1.0 * [OA] + 3.3 * [BC]
\end{aligned} \tag{1}
$$




$$b_{sct_{sum}} = 4.6 * f_{RH} * [(NH_4)_2SO_4] + 5.5 * f_{RH} * [(NH_4)NO_3] +$$
$$5.0 * f_{RH} * [(NH_4)Cl] + 1.8 * [OA] + 8.8 * [BC] \qquad (2)$$

An optimisation approach is then used to estimate the contribution of each species to the measured lidar extinction at each 30 m layer (up to a height of 6 km) given the empirical relationships derived in Equations 1 and 2. In this case the assumption

is made that the MSE and MAE values for each species hold with height. Relative humidity is obtained for the column from the European Centre for Medium-range Weather Forecast (ECMWF) ERA5 dataset (ECMWF, 2018) and is mapped onto the lidar height levels in order to estimate the change in the $f_{RH}$ with height and thus the effects of hygroscopic growth on the inorganic aerosol species. As the lidar operates at 532 nm and the PAX optical properties are measured at 870 nm, the lidar extinction is scaled to the PAX wavelength using the Angstrom exponent from a nearby Aerosol Robotic Network (AERONET) station

(Beijing CAMS) before the optimisation. Overall, this provides vertical profiles of extinction attributed to each aerosol species which are then converted to aerosol optical depth (AOD) by integrating over each 30 m layer.

## 2.4   Model Description and setup

Fast-JX is an interactive photolysis scheme designed to efficiently and accurately calculate photolysis rates for use in global atmospheric models at minimal computational cost (Wild et al., 2000; Bian and Prather, 2002; Neu et al., 2007; Prather, 2015).

The scheme apportions light from wavelengths 177 to 850 nm into 18 bins to permit calculation of photolysis rates appropriate to both tropospheric and stratospheric chemistry. Cloud and aerosol optical depths are used along with the scattering phase functions for appropriate particle types to solve the 8-stream multiple scattering problem (Wild et al., 2000). This allows calculation of the photolytic intensity which can be used to determine photolysis rate coefficients for key atmospheric species (e.g. J[O$^1$D] and J[NO$_2$]). The scheme also utilises a quadrature approach to account for multiple layers of overlapping clouds

(Prather, 2015).

The Fast-JX scheme is run here in 'stand-alone' mode using offline data rather than run interactively within a chemical transport model (CTM) framework. This allows it to be constrained using observations where these are available. In this study, the optical depths for aerosols from the lidar chemical apportionment are used in conjunction with data on cloud (cloud cover, liquid water content and ice water content) and meteorological variables (temperature and relative humidity) from the

ERA5 reanalysis dataset (ECMWF, 2018). Atmospheric columns of O$_3$ and NO$_2$ are provided from the ERA5 and the CAMS reanalysis datasets (CAMS, 2018; Inness et al., 2018) respectively, to account for absorption from gas-phase species. Due to high concentrations of NO$_2$ in Beijing, particularly in the boundary layer, the standard version of the Fast-JX code was modified to account for the significant attenuation that can occur when photons are absorbed by NO$_2$. In addition, the scattering phase function for BC was updated to account for the ageing and coating of BC particles. This is done using the BC core size and

coating thickness determined from SP2 measurements to derive asymmetry parameters at each Fast-JX wavelength. These asymmetry parameters were used to estimate the first eight terms of the scattering phase function using a Legendre expansion of the Henyey-Greenstein function (Henyey and Greenstein, 1941).



## 3   Results

In order to critically evaluate the impact of haze pollution events on photolysis rates during the winter and summer campaign periods, a range of different scenarios are run (Table 1). These include runs where the radiative effects of each key aerosol species and clouds are switched on in isolation. This enables the contributions of clouds and aerosols to be determined, and

for the contribution of key aerosol species to be identified respectively. The scenario where both cloud and aerosol effects are turned on represents the best model simulation of photolysis rate constants during the campaigns and allows critical evaluation of the model against observed values of J[O$^1$D] and J[NO$_2$].

### 3.1   Fast-JX evaluation

Observed J[O$^1$D] and J[NO$_2$] are captured well by the model during both the winter and summer periods, as shown in Figure 1.

In general, when the sun is highest in the sky at local noon, Fast-JX tends to capture observed J[O$^1$D] within 8 % (3.7 % in summer and 7.8 % in winter, averaged over all days), marginally higher than observations in both cases. For J[NO$_2$] the model performance is much better during the summer campaign (positive bias of 5.6 %) than the winter (bias of 20.4 %). In the summer, on clear sky days (28$^{th}$ May, 1$^{st}$, 7$^{th}$, 9$^{th}$, 14$^{th}$, 15$^{th}$ and 16$^{th}$ June) the model performs slightly less well for J[O$^1$D] (5.5 % average bias) and slightly better for J[NO$_2$] (4.7 % bias). In contrast, on winter clear sky days (19$^{th}$, 22$^{nd}$, 27$^{th}$

November and 1$^{st}$ December) the model performs better for J[O$^1$D] (2.7 % bias) but less well for J[NO$_2$] (23.1 % bias). These discrepancies may be linked to uncertainties in the retrieval of AOD from the lidar instrument and therefore to underestimation of the background aerosol on these less polluted days. In addition, errors in the column O$_3$ from the ERA5 reanalysis could be responsible for the positive bias of model J[O$^1$D]. However, the ERA5 column O$_3$ was independently validated against Brewer measurements over Beijing which indicated that the total column was captured well by the reanalysis product (mean bias of

-2.0 % for summer and 3.4 % for winter), indicating that this is likely to be a smaller source of error. There are also issues with stray light during calibration of the spectrometer instrument at shorter wavelengths (below 300 nm) which may affect the fluxes derived in this part of the spectrum and thus the measured photolysis rates, particularly for J[O$^1$D]. Overall, however, the model is shown to perform reasonably well, capturing the magnitude of the reductions in photolysis rates observed during severe haze episodes in both winter (16$^{th}$–18$^{th}$, 20$^{th}$ and 29$^{th}$ November) and summer periods (6$^{th}$, 22$^{nd}$ and 23$^{rd}$ June). This

is particularly important during the summer where photochemistry is most active and large reductions in photolysis rates can have important impacts on oxidant concentrations and the production and destruction of key pollutants (e.g. O$_3$ and secondary organic aerosol formation).

### 3.2   Chemical apportionment of AOD

Figure 2 shows the vertical profiles and contributions to total column AOD for both campaigns as derived from the optimisation

approach descibed in Section 2.3. The differences between haze periods and non-haze periods are also highlighted. Haze is defined here as conditions where the PM$_{2.5}$ concentration is larger than 75 $\mu$gm$^{-3}$ (corresponding to an air quality index, AQI, of 100) which is the daily air quality limit for China (Shi et al., 2018). In winter (NH$_4$)$_2$SO$_4$ (39.1 %) and BC (30.5 %)





provide the largest contributions to total column AOD during haze periods with $NH_4NO_3$ and organic aerosol making similar contributions (13.1 % and 12.3 % respectively) and $NH_4Cl$ making the smallest contribution (5.0 %). Vertical AOD profiles show high values in the boundary layer below 1 km which then decline rapidly with altitude before peaking again above 3 km where there is evidence of an elevated pollution layer (EPL). This phenomenon has been demonstrated previously in

stable conditions during haze episodes when the topography around Beijing can lead to the Mountain Chimney effect where pollutants build up in elevated layers rather than disperse through the free troposphere (Chen et al., 2009; Liu et al., 2018b). During cleaner winter periods $(NH_4)_2SO_4$ (29.8 %) and BC (23.2 %) still dominate total column AOD and $NH_4NO_3$ and organic aerosol make very similar contributions to those during haze periods (12.9 % and 14.3 %). However, in contrast to haze periods, $NH_4Cl$ makes the third biggest contribution at 19.8 %. As these cleaner periods are charaterised by lower levels

of particulates and more unstable conditions, AOD values are in general much lower than in hazy conditions and the largest values are seen within the boundary layer.

During the summer campaign, in both haze and cleaner periods, organic matter dominates the contribution to total column AOD (37.9 % and 44.2 % respectively) with $(NH_4)_2SO_4$ providing the second largest contribution (21.0 % and 15.5 % respectively). BC is shown to contribute less to AOD during haze days (6.6 %) than in cleaner periods (12.7 %). In contrast to the

winter, the highest AOD values in hazy periods are seen in an EPL which lies between 3–5 km and is dominated by BC and organic matter, although all other species are also elevated. Although haze conditions occur much less frequently in the summer campaign, the AOD values for BC and organic matter at these higher levels are comparable to those seen in the winter months, with peaks of around 0.015. During the summer campaign, all species except organic matter show much lower AOD values during the cleaner periods than during haze, and there is little indication of an EPL layer. There is substantial organic matter

still present in these cleaner periods, and this is maximum at 2–3 km where AOD values peak at around 0.006 (higher than during cleaner periods in the winter campaign). Finally, to check consistency, the total column optical depth from the lidar was compared to AERONET values at a nearby site (Beijing CAMS) showing good agreement for both campaigns.

### 3.3    Impacts of aerosol species on photolysis rates in haze conditions

As Fast-JX is run in offline mode, the effects of each aerosol species can be determined independently, and the impacts under

haze conditions are shown in Figure 3. This allows quantification of the impacts of each species on photolysis rates during haze episodes in Beijing. During winter, absorption by BC has the greatest impact on both $J[O^1D]$ and $J[NO_2]$, resulting in 23.8 % and 23.1 % reductions in the lowest 1 km respectively, compared to clear sky conditions. The impact of BC reduces with height, but continues to show the largest response up to 4 km for $J[O^1D]$ (-5.7 %) and 3 km for $J[NO_2]$ (-6.3 %). At higher altitudes the effects of scattering aerosol begin to dominate, with a general enhancement in photolysis rates compared to clear

sky conditions of 1.3–3.8 % for $J[O^1D]$ and 1.5–6.8 % for $J[NO_2]$. The most pronounced effects are seen towards the top of the lidar column (6 km), where there is substnatial backscattered solar radiation from the polluted boundary layer below. The largest increases are due to $(NH_4)_2SO_4$ (3.8 % for $J[O^1D]$ and 6.8 % for $J[NO_2]$) which corresponds to the presence of large amount of $(NH_4)_2SO_4$ in the EPL during haze periods in winter (Figure 2). In the surface layer, below the elevated levels of




aerosol, the scattering aerosol lead to reductions of -1.7 % to -4.4 % for J[O$^1$D] and -3.4 % to -7.0 % for J[NO$_2$], with OA producing the largest reduction in both cases.

In contrast, during the summer, scattering by OA dominates the response of both J[O$^1$D] and J[NO$_2$]. At the surface, OA leads to a 2.9 % reduction in J[O$^1$D] and a 2.4 % reduction in J[NO$_2$]. Higher up the column, within and above the EPL layer,
scattering aerosol leads to increases in photolysis for both species, with the effects of OA producing the dominant response (3.7–8.4 % for J[O$^1$D] and 4.3–10.1 % for J[NO$_2$]). This represents the high levels of backscatter from the EPL layer during haze events in the summer campaign. Overall, the relative impacts of the scattering aerosol compared with clear sky conditions are similar in winter and summer, but due to the higher rates of photolysis in summer the absolute impacts are much larger. Absorption by BC is still evident during the summer leading to reductions in photolysis of 0.1–2.4 % for J[O$^1$D] and 1.0–3.8 %
for J[NO$_2$]. However, in contrast to winter, the impacts of BC are lower than those of OA in all layers with exception of J[NO$_2$] in the surface layer.

### 3.4  Diurnal impacts of aerosols on photolysis rates

Figure 4 shows the impacts of all aerosol on the vertical profiles of J[O$^1$D] and J[NO$_2$] for each daylight hour averaged over each campaign period. During the winter, the strong effect of absorption by BC is evident for both J[O$^1$D] and J[NO$_2$] with
reductions in excess of 20.0 % in the lowest 500 m and reaching maximum reductions of 34.0 % for J[O$^1$D] and 40.4 % for J[NO$_2$]. Reductions in J[NO$_2$] show a stronger diurnal pattern over the day, with the greatest effects around sunrise, where reductions of as much as 20.0 % are seen up to an altitude of 5 km. This is when the atmospheric path is at its longest and absorbing species such as BC have a more dominant effect. Approaching noon the effects of scattering aerosol start to dominate down through the column and this results in increases in J[NO$_2$] to a maximum of 10.5 % at 6 km altitude. For J[O$^1$D] the effects
of scattering are less pronounced and absorption continues to dominate below 3 km. The effects of scattering are more evident above this level, and J[O$^1$D] is enhanced by around 6.0 % during the middle of the day.

During the summer, the responses of J[O$^1$D] and J[NO$_2$] are similar, with the absorption effect of BC evident in a very shallow surface layer about 200–300 m deep. The exception to this is around sunrise and sunset where reductions in photolysis rates extend to altitudes of 2–3 km. In this layer, reductions in photolysis rates for both species are in excess of 20.0 %, and
these are most pronounced for J[NO$_2$] where reductions reach 66.0 %. The effects of scattering aerosol are more pronounced during the summer than the winter and are seen much lower in the column, particularly during the middle of the day. The impacts on J[NO$_2$] are slightly greater than for J[O$^1$D], with increases of 10.0–15.0 % compared with 5.0–10.0 % at 1–6 km altitude. Although these increases are not as large as the reductions seen in the boundary layer, they have the potential to have significant impacts on chemical processes through influences on oxidant concentrations. This can have further impacts on
lifetimes of other pollutants in the free troposphere. These findings also allow quantification of the full impacts of haze pollution throughout the lower troposphere and highlight the non-local impacts of aerosols. That is that particulate matter confined mainly to the boundary layer is shown to produce significant impacts at altitude. This extends findings of previous studies that largely focus on the impacts of haze pollution on surface photolysis rates and not those throughout the free troposphere (Li et al., 2005; Xing et al., 2017).



### 3.5 Cloud vs aerosol impacts during campaign periods

The average impacts of clouds and aerosol on J[O$^1$D] and J[NO$_2$] are shown for both campaigns for haze days in Figure 5. During the winter, aerosols produce reductions of more than 30.0 % at the surface for both photolysis rates. The impact of clouds in this lowest layer is much smaller, with reductions of around 7 % for both species. At higher altitudes, the effects of aerosol are less dominant and backscatter from clouds is more evident, leading to increases in photolysis rates for both species. The largest increases are seen in the 1–2 km layer where photolysis rates increase by 46.4–57.5 % with the impacts for J[NO$_2$] slightly larger than for J[O$^1$D]. The effects of cloud backscatter are smaller higher up the column, and increases of 17.0–31.9 % are seen in the 5–6 km layer, with largest impacts again seen for J[NO$_2$]. This pattern is reflected in the combined impact of cloud and aerosol which shows 33.3–34.4 % reductions in the surface layer, 32.0–41.7 % increases in the 1–2 km layer, and smaller 19.9–39.3 % increases in the 5–6 km layer.

In summer, during haze conditions, clouds produce the largest impact on photolysis rates in the surface layer, with approximately twice the impact of aerosols (reductions of 11.3–10.2 % compared to ≈6.1 %). The effects of high levels of scattering aerosol (particularly OA) during summer haze conditions are evident higher in the column where the increases in photolysis rates due to aerosol are much larger than that from clouds. This is reflected in the combined effects of cloud and aerosols which see reductions in photolysis rates in the lowest 3 km (0.1–17.2 % for J[O$^1$D] and 1.2–15.7 % J[NO$_2$]) switching to increases from 3–6 km (8.8–13.7 % for J[O$^1$D] and 11.3–17.8 % J[NO$_2$]) where the backscatter from aerosol dominates over that from clouds.

When the impacts are averaged over all days in each campaign period, an interesting picture emerges. In the winter, aerosol produce the largest impacts in the surface layer (-15.9 % for J[O$^1$D] and -17.3 % J[NO$_2$]), with the effects of cloud more dominant at higher altitudes (14.8–26.2 % for J[O$^1$D] and 25.6–34.7 % J[NO$_2$]). This shows that background levels of aerosol during the winter, even at levels below that classified as haze, are sufficiently high to produce a greater impact on photolysis than clouds. As shown in Figure 2, a polluted layer consisting mainly of BC and (NH$_4$)$_2$SO$_4$ is present in the lowest km even on non-haze days, and this most likely contributes to the dominant effect of aerosol at the surface. For the summer campaign, averaged over all days, cloud impacts dominate over aerosol impacts throughout the column for both species. However, the presence of an elevated layer of OA (Figure 2) at 2–3 km is evident on the non-haze days where aerosol and clouds produce comparable increases in photolysis rates ( 5.0 % for J[O$^1$D] and  7.0 % for J[NO$_2$]).

## 4 Discussion

This study presents an in-depth investigation into the impacts of haze pollution on photolysis rates during two intensive field campaigns (Winter 2016 and Summer 2017) in Beijing. For the first time, an observation driven approach is used to quantify how different aerosol species contribute to changes in photolysis rates and to explore how they influence photochemistry during haze events in a megacity.

On haze days, in the winter and summer campaign periods, aerosols show distinct and contrasting impacts on J[O$^1$D] and J[NO$_2$]. The effects of absorbing species such as BC dominate during the winter leading to large reductions in J[O$^1$D]



and J[NO$_2$] in the lowest 1 km of 23.8 % and 23.1 % respectively. During the summer scattering aerosol dominates with OA producing the largest response throughout the column leading to reductions of 2.9–2.4 % in the lowest 1 km and increases of 10.0 % at 3–4 km. These differences largely reflect the different pollution sources during the campaign periods with high levels of coal burning during the winter season leading to large emissions of soot (BC) and high production of (NH$_4$)$_2$SO$_4$

particles. During the summer, emissions of biogenic volatile organic compounds (bVOCs) are much larger leading to formation of secondary organic aerosol (Mentel et al., 2013; Riipinen et al., 2011) which can account for the dominance of OA in this season.

To test the sensitivity of the results to assumptions about the mixing state of aerosol, an additional simulation was performed treating BC as externally mixed rather than internally mixed, following Liao et al. (1999). With externally mixed BC, the effects

of the absorbing species still dominate the response of photolysis rates during the winter campaign, see Figure 3. However, the magnitudes of reductions in the lowest 1 km for both J[O$^1$D] and J[NO$_2$] is larger (44.0 % and 45.0 % respectively) than when the BC is assumed to be coated (23.8 % and 23.1 % respectively). In the summer, the effect of externally mixed BC leads to a slightly larger contribution from BC to reductions in J[O$^1$D] and J[NO$_2$] throughout the column, although OA still provides the dominant response for both species (Figure 3).

It is also demonstrated that despite particulate levels being lower on average during the summer months, haze events where the AQI is higher than 100 still occur. On haze days the mean relative impacts of aerosols on photolysis rates is lower in summer than in winter, with reductions in J[O$^1$D] of just 6.1 % in the surface layer compared to 28.5 % in winter. However, summer J[O$^1$D] values in this layer are an order of magnitude higher than in winter (1.6x10$^{-5}$s$^{-1}$ compared to 2.6x10$^{-6}$s$^{-1}$) and so the absolute changes are higher in summer (1.0x10$^{-6}$s$^{-1}$ compared to 7.4x10$^{-7}$s$^{-1}$). This is important as incoming solar radiation is

at its highest during the summer months and is often the 'photochemical limitation' in the formation of O$_3$ (Tie and Cao, 2009). During the summer, these large reductions in surface J[O$^1$D] and J[NO$_2$] could lead to lower O$_3$ levels than if particulate levels were low. As demonstrated by Li et al. (2011), the probability of O$_3$ peaks greater than 120 ppbv increased dramatically with the removal of aerosol from their simulations, and the response of OH concentrations was shown to be approximately linear to changes in J[O$^1$D]. Therefore, if controls were implemented to reduce aerosol concentrations, the subsequent increases in

J[O$^1$D] at the surface could lead to enhanced O$_3$ concentrations, a major problem in Beijing where summer O$_3$ is already very high (Wang et al., 2006; Xue et al., 2014; Ni et al., 2018). Enhanced near-surface photolysis rates would also increase O$_3$ production via NO$_2$ photolysis, however this would be balanced by the equivalent rise in NO from NO$_2$ photolysis. Furthermore, the enhancement of J[O$^1$D] would increase OH concentrations which would subsequently increase HO$_2$ and RO$_2$ and lead to a net rise in O$_3$ concentrations. In the winter, a similar response would be expected, although due to lower photolysis rates and

much lower O$_3$ concentrations, the effects of particulate control strategies would have a lesser effect on oxidant concentrations.

To evaluate the potential photochemical impact on oxidants, a simple experiment was performed using a photochemical box model incorporating the generic reaction set for volatile organic compound oxidation (Topping et al., 2018). The response of O$_3$ concentrations to the presence of aerosols is similar to that of J[O$^1$D] in both seasons with the biggest reductions (12.0 % or larger) seen in the lowest 500 m, see Figure 6. These reductions are more pronounced throughout the day in the winter than

in the summer months. The effect of scattering aerosol can be seen further up the column, where enhanced photolysis rates due

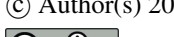


to aerosol result in increases in $O_3$ concentrations of 3.0–6.0 %. As with the photolysis rates, the largest impacts are seen in the middle of the day and lower down the column during the summer. The impacts on OH largely match those on $O_3$, but the responses are much smaller, with 0.0–3.0 % decrease at the surface in winter and a similar magnitude increase higher up the column in the summer.

The implication of this is that if aerosols were not present surface $O_3$ and OH concentrations would in fact be higher by around 12.0 % and 3.0 % respectively. In contrast, removal of aerosols would see drops in $O_3$ higher up the column, particularly in the summer due to the removal of scattering aerosol and the resultant reductions in $J[O^1D]$ and $J[NO_2]$ (see above). In the winter, when $O_3$ are much lower this may have little impact, but during the summer months when daytime concentrations can exceed 100 ppbv, this could see a potential shift towards more frequent $O_3$ pollution episodes. It should be noted here

however, that in this idealised box model scheme, the photolysis of HONO is not included. HONO photolysis occurs as similar wavelengths to that of $NO_2$ and is an important source of OH in urban environments. Therefore chemical impacts simulated here, particularly for OH, may be underestimated. Therefore, a much more detailed treatment of gas-phase and heteorogeneous chemistry in a box model or regional air quality model is needed to fully quantify the impact of these photolysis rate changes on urban oxidant concentrations under haze conditions.

The diurnal pattern of haze impacts on photolysis is shown to differ between winter and summer (Figure 4) with the reductions in $J[O^1D]$ and $J[NO_2]$ extending further up through the column during winter. This reflects the much deeper haze layers at the surface throughout the winter (Figure 2). In the summer months, during the middle part of the day, significant reductions in photolysis occur near the surface but at higher altitudes aerosols result in increases of between 6.0 and 16.0 % depending on the species. Therefore, any reductions in aerosol concentrations during the summer months and thus reduction in this scattering

effect (particularly from OA), would reduce $J[O^1D]$ and $J[NO_2]$ higher up in the column. As discussed above, lower values of $J[O^1D]$ and $J[NO_2]$ could reduce $O_3$ and OH concentrations, and thus potentially have important implications for the oxidation capacity in the free troposphere.

Previous work has shown that in air masses influenced by urban pollution, the impacts of aerosol on photolysis rates are greater than the influence of cloud cover (Tang et al., 2003). This study shows that during moderate to severe haze events

in Beijing, which occur in both summer and winter, the response of photolysis to aerosol often dominates over cloud with contrasting results. In the winter absorbing aerosol dominates over cloud at the surface leading to 28.5–30.9 % reductions in photolysis rates compared to just 6.6–7.4 % reductions from the presence of clouds. In the summer, cloud impacts dominate near the surface with the scattering effect from an elevated layer of predominantly OA (Figure 2) taking over at around 3–4 km altitude where photolysis rates are increased by 12.0–13.7 % compared to very minor decreases from cloud (0.24–0.32 %).

These patterns are in agreement with previous studies which also demonstrate dominant responses to aerosol in urban air plumes (Liao et al., 1999; Tang et al., 2003). In addition, this work agrees with that of Liao et al. (1999) that the presence of a cloud layer acts to accentuate the reduction in surface photolysis rates for the two reactions considered here. Furthermore, in winter in particular, for the average impacts over all days in the campaign period, the aerosol effects (15.9–17.3 % reductions) are roughly double the cloud impacts (7.9–8.1 % reductions) in the surface layer. This highlights despite the very episodic nature of haze events in Beijing (Figure 1), background aerosol concentrations are still sufficiently high enough at the surface





to produce significant reductions in photolysis rate constants. This is likely due to reasonably high AOD values attributed to BC and $(NH_4)_2SO_4$ on the cleaner days (Figure 2). It should be noted that this response is representative for the APHH winter campaign only and these respective effects of cloud and aerosol species are likely to vary greatly between different winter seasons. The results presented in this study however emphasise the significant impact particulate pollution can have on

photochemistry.

The results presented here are at the lower end of previous estimates on the impacts of aerosol on photolysis rates (Liao et al., 1999; Tang et al., 2003; Li et al., 2011). For example, Tang et al. (2003) estimate 29.2–38.5 % reductions in J[O$^1$D] in the lowest 3 km whereas Li et al. (2011) present reductions of 37.2–53.3 %. This study estimates 5.4–30.8 % reductions in the winter and 6.1 % reductions to 8.8 % increases in the lowest 3 km in the summer. These studies either use limited aircraft data

AOD (Tang et al., 2003) or total column AOD data (from a combination of satellite and ground based instruments (Li et al., 2011)) to verify the aerosol fields and thus were unable to fully verify the accuracy of the vertical distributions of different aerosol species throughout the column. The results presented in this work greatly extend these earlier estimates by using a combined statistical and observation driven approach to derive chemically apportioned aerosol extinction from lidar data. This allows more accurate constraint of the vertical distribution of aerosol to be made, including which species provide the largest

contribution to extinction throughout the column. As discussed above, this results in layers of haze near to the surface ($\approx$3 km in winter and $\approx$1 km in summer) dominated by absorbing species in the winter (BC) and scattering species in the summer (OA). This leads to large reductions in J[O$^1$D] and J[NO$_2$] at the surface and significant increases at altitude during both campaigns. These increases at height are more emphasised during the summer due to the backscatter from the layer of OA present at the surface (Figure 2). This accounts for the difference in patterns observed between this study and the previous estimates

by Tang et al. (2003); Li et al. (2011) and highlights the sensitivity of the responses in photolysis rates not only to the vertical distribution of aerosol but also the chemical speciation of the particulates. The observation constrained approach presented here provides the best possible estimate of both allowing more accurate quantification of the impacts of haze pollution on photolysis rates. Running the Fast-JX code in 'offline' mode also enables the relative impacts of clouds and different aerosol species to be quantified independently. This allows a critical evaluation of whether clouds or a particular aerosol species is causing the

greatest photochemical impact during a specific pollution episode to gain insight as where to target particulate matter control strategies.

## 5   Conclusions

This study presents, for the first time, application of aerosol composition data and lidar extinction profiles to drive the Fast-JX photolysis model in order to quantify the effects of severe haze pollution on J[O$^1$D] and J[NO$_2$] over Beijing during two

intensive measurement campaign periods (winter 2016 and summer 2017). The model is shown to capture observed J[O$^1$D] and J[NO$_2$] well during both campaigns, particularly the reductions that occur during haze events (AQI > 100). Such episodes occur during both campaign periods and lead to reductions in surface J[O$^1$D] of 27.4 % (summer) to 33.7 % (winter) and reductions





of 40.4 % (winter) to 66.2 % (summer) in $J[NO_2]$. Despite much lower particulate concentrations in the summer campaign, the absolute changes in photolysis rates are shown to be larger than in winter for both $O_3$ and $NO_2$.

In the winter, absorbing aerosols such as BC are shown to dominate the photolysis response to aerosol leading to mean reductions of 23.8 % and 23.1 % respectively for $J[O^1D]$ and $J[NO_2]$ in the lowest 1 km. In contrast, in the summer, scattering

aerosols such as OA dominate the response leading to mean decreases of around 2.4–3.8 % at the surface and increases of 8.4–10.1 % at higher altitudes (3–4 km). During haze episodes, these effects often dominate over those attributed to cloud cover. This emphasises that in heavily polluted urban environments such as Beijing, aerosols are present in such large concentrations that they produce larger impacts on photolysis than naturally occurring phenomena such as clouds, and are likely to have other important chemical and dynamical impacts on the urban environment.

The results presented here greatly improve on previous studies by using a combined statistical and observational driven approach to provide a more accurate representation of the vertical distribution and speciation of aerosol extinction. This allows a more accurate constraint on the estimated impacts of haze pollution on photochemistry. Critically, this shows which aerosol species is causing the largest photochemical impact during each campaign period and therefore allowing the identification of which source sectors to target particulate control strategies on. For example during the APHH winter campaign, sources of BC

(e.g. power generation, residential heating) contribute the most to photochemical impacts during haze events. Such strategies would not only reduce pollutant concentrations but would also offset the photochemical impacts demonstrated here. Further-more, the non local impact of potential emissions and pollution control strategies is emphasised. Emissions cuts implemented at the surface not only produce large impacts in surface oxidant concentrations but also result in significant impacts in the free troposphere. Using an idealised photochemical box model, if particulates were completely removed during the APHH

campaigns, surface $O_3$ concentrations could be enhanced by around 12.0 % (3.0 % for OH). In contrast, particulate controls could reduce $O_3$ by 3.0–6.0 % in the free troposphere. However, any control policy is also likely to impact concentrations of other pollutants such as $NO_x$ and VOCs and therefore due to the complex non-linearities involved, the response of atmospheric oxidants is likely to vary dependent on the magnitudes of the emissions changes. To fully quantify the effects of pollutant haze on the urban atmospheric oxidation capacity, a more detailed air quality model study is needed that fully incorporates the

optical properties of urban aerosol and their impacts on photolysis rates demonstrated here along with treatment of gas-phase and heterogeneous photochemistry and urban meteorology.

*Data availability.*    The data generated in this study will be made available through the CEDA data archive at http://dx.doi.org/10.XXXXX/XXXXXX.

*Author contributions.*    MH and OW conceived this study. MH performed the data analysis and ran model simulations. TY collected and provided the lidar data at the IAP tower site. YS, WX and CX collected and provided the aerosol composition data and optical properties

measured at the tower site. LW, ES and DH collected and provided the observed photolysis rates. DL collected and provided the optical properties for black carbon from the SP2 instrument. MH and OW wrote the manuscript with input from all authors.



*Competing interests.* The authors declare that they have no conflict of interest.

*Acknowledgements.* This authors acknowledge funding from the AIRPRO (An Integrated Study of Air Pollution PROcesses in Beijing) project, a Newton Innovation Fund project funded by the UK Natural Environment Research Council (Grant Numbers NE/N006925/1, NE/N006895/1 and NE/N007123/1), the Strategic Priority Research Program of the Chinese Academy of Sciences 'Beautiful China' project
5 (Grant Number XDA19040203) and the Natural Science Foundation of China (Grant Number 41571130034). ES acknowledges funding from the SPHERES NERC Doctoral Training Programme.



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



**Table 1.** Model scenarios run during both campaign periods. The column headers indicate whether the radiative effects of each aerosol species/cloud is turned on in the respective scenario

| Scenario | Cloud | Aerosol (all) | $SO_4^{2-}$ | $NO_3^-$ | $Cl^-$ | OA | BC |
|---|---|---|---|---|---|---|---|
| Clear Sky | Off | Off | Off | Off | Off | Off | Off |
| Cloud Only | **On** | Off | Off | Off | Off | Off | Off |
| Aerosol Only | Off | **On** | **On** | **On** | **On** | **On** | **On** |
| SO4 Only | Off | Off | **On** | Off | Off | Off | Off |
| NO3 Only | Off | Off | Off | **On** | Off | Off | Off |
| CHL Only | Off | Off | Off | Off | **On** | Off | Off |
| ORG Only | Off | Off | Off | Off | Off | **On** | Off |
| BC Only | Off | Off | Off | Off | Off | Off | **On** |
| Aerosol + Cloud | **On** | **On** | **On** | **On** | **On** | **On** | **On** |





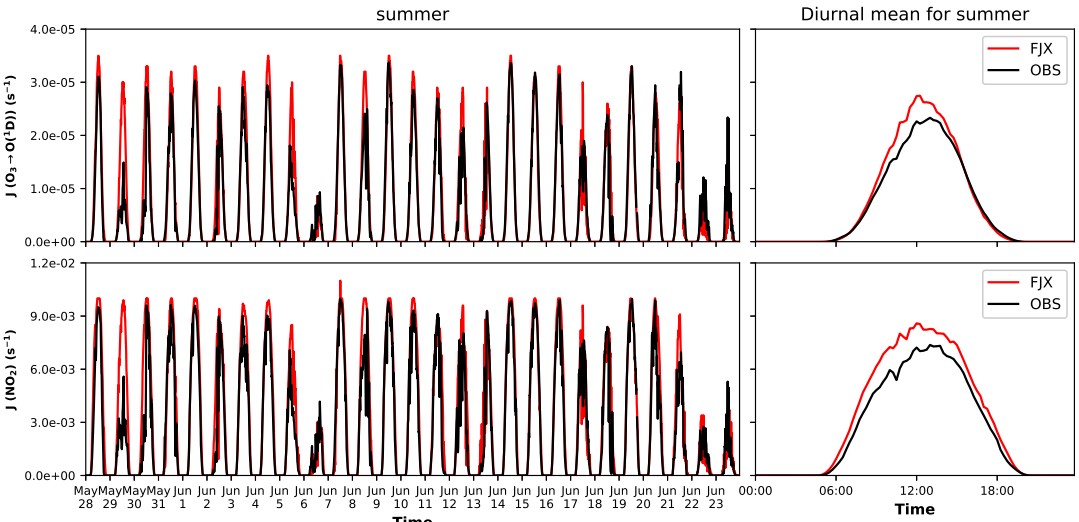

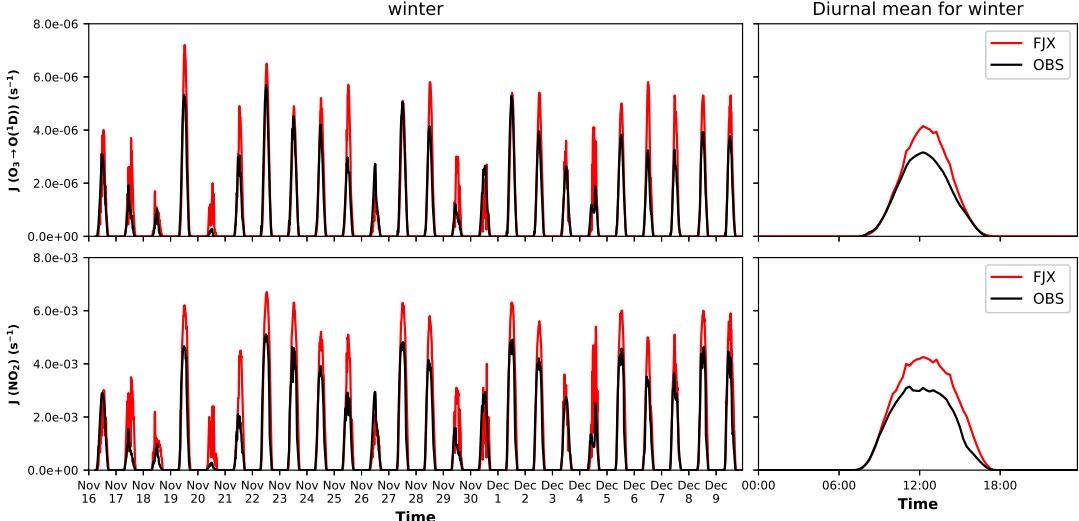

**Figure 1.** Modelled versus observed photolysis rate constants for the winter (top panel) and summer (bottom panel) campaign periods. Time series (left hand plots) and campaign diurnal averages (right hand plots) of J[O$^1$D] and J[NO$_2$] are shown for simulations with Fast-JX (FJX) including cloud and aerosol (red lines) and for observations. Note the difference in magnitude of rate constants between summer and winter.







**Figure 2.** Contributions to total column AOD (top plots) and vertical profiles of AOD (lower plots) for each aerosol component as derived from the optimisation approach. Data are presented for for haze and non-haze periods (see text for definition) in both campaigns from the surface up to 6 km altitude.




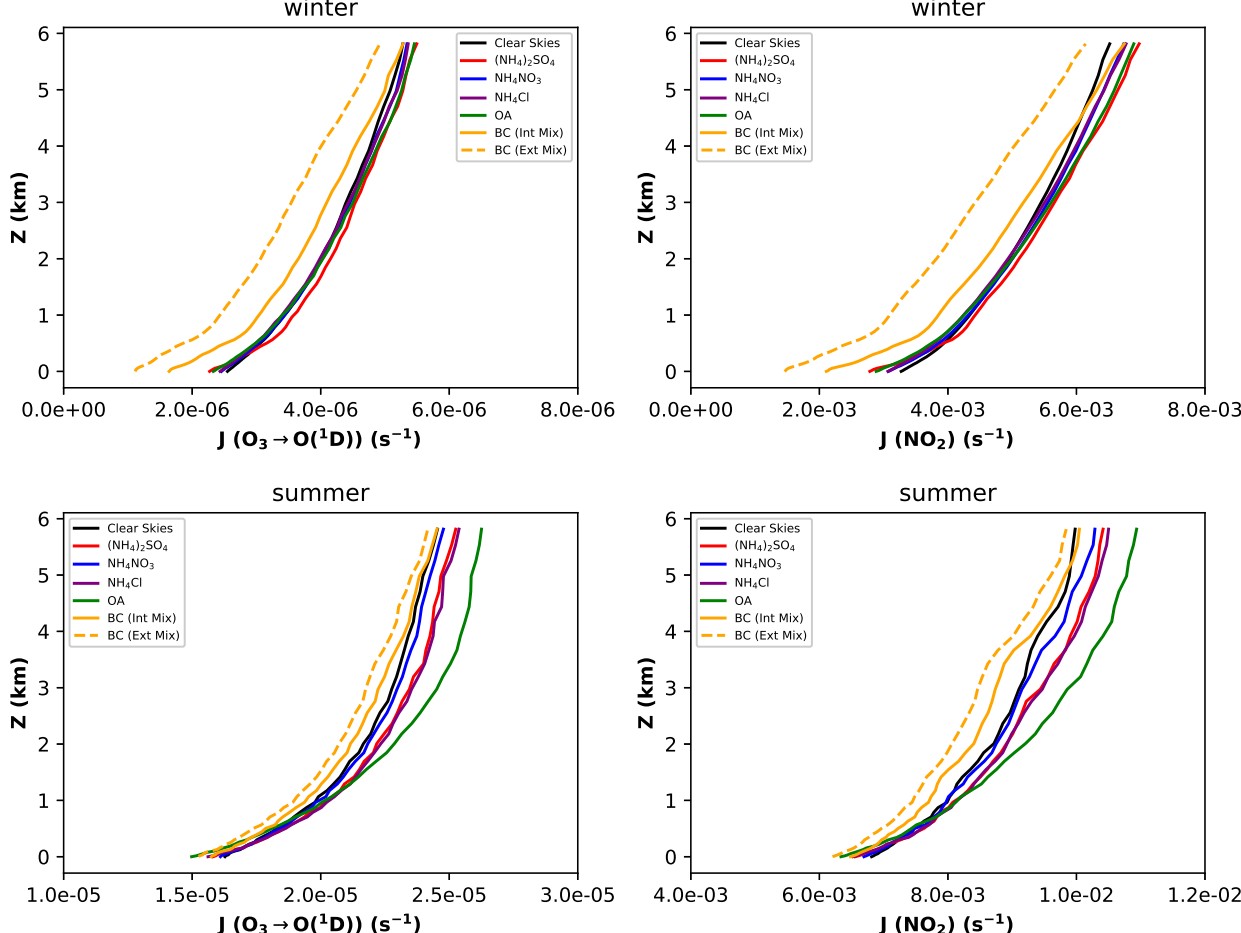

**Figure 3.** Vertical profiles of J[O$^1$D] (left) and J[NO$_2$] (right) for simulations where the radiative effects of each aerosol component is included in isolation. Profiles are shown for the winter (top) and summer campaigns (bottom) and show impacts of OA (solid green), (NH$_4$)$_2$SO$_4$ (solid red), NH$_4$NO$_3$ (solid blue), NH$_4$Cl (solid purple) and coated BC (solid orange) and respectively. Photolysis rate constants under clear sky conditions (solid black line) are shown for reference. The radiative impacts of BC when it is assumed to be externally mixed rather than coated is shown for comparison (dashed orange line).





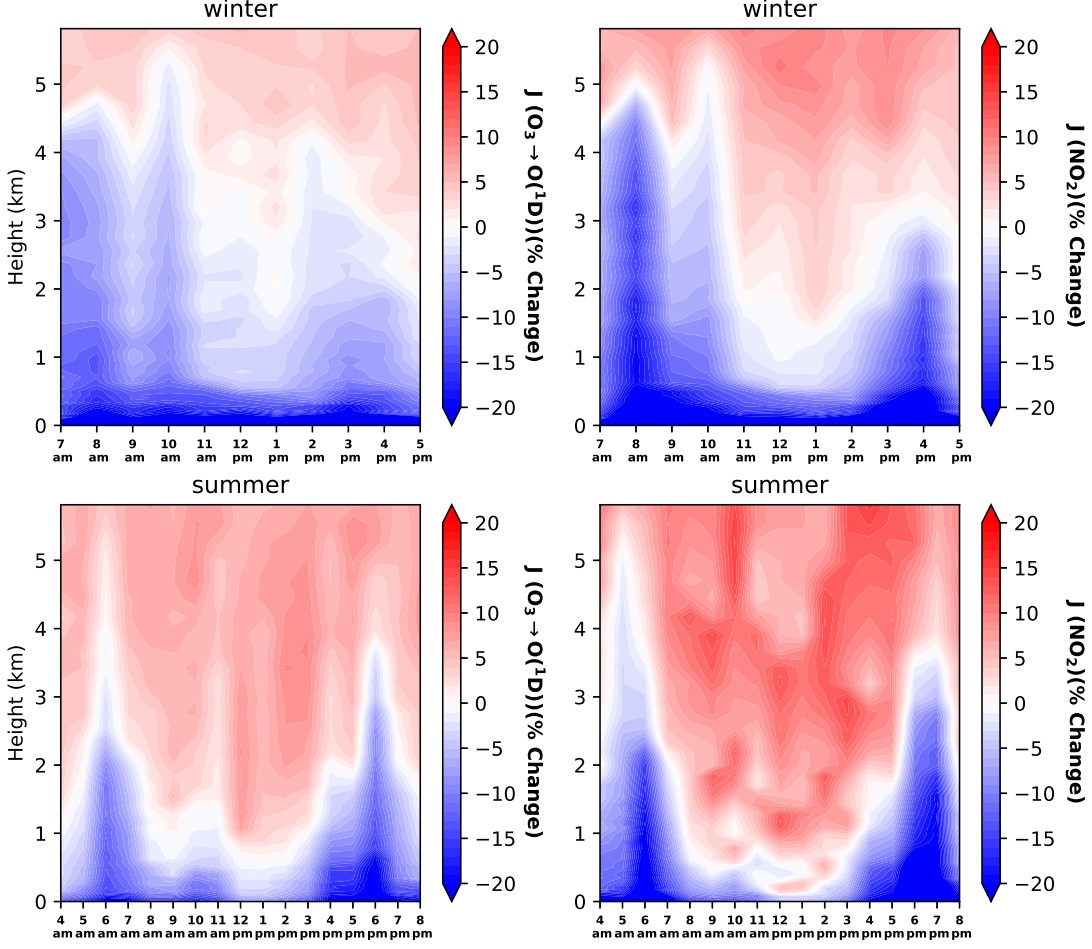

**Figure 4.** Diurnal profiles of aerosol impacts on J[O$^1$D] (left column) and J[NO$_2$] (right column) for winter (top row) and summer (bottom row) campaign periods. Plots show the mean relative difference with respect to clear sky conditions, with blue representing a reduction due to aerosols and red representing an increase. Note the difference in time scales between seasons which reflects longer daylight hours in summer.



**Figure 5.** Mean relative impacts of aerosols (red bars), clouds (grey bars) and combined cloud and aerosol (blue bars) on J[O$^1$D] (left column) and J[NO$_2$] (right column) during the winter (top row) and summer (bottom row) campaign periods. The relative differences are with respect to clear sky conditions and are calculated as the average for periods classified as haze, in 1 km layers from the surface to 6 km.





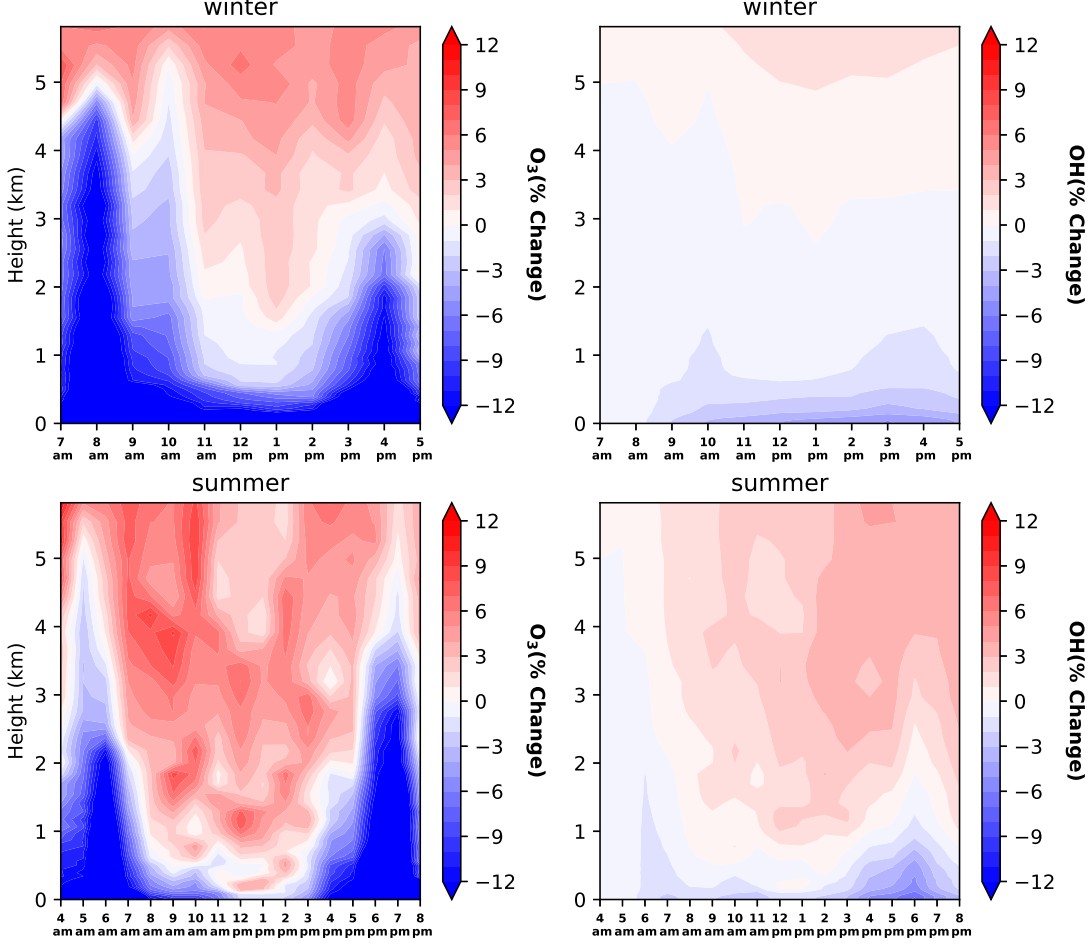

**Figure 6.** Diurnal profiles of aerosol impacts on O$_3$ (left column) and OH (right column) concentrations for winter (top row) and summer (bottom row) campaign periods. These changes are calculated using a simple chemical box model, and plots show the mean relative difference with respect to clear sky conditions, with blue representing a reduction due to aerosols and red representing an increase.