# Peer review of "Photochemical impacts of haze pollution in an urban environment"

_Atmospheric Chemistry and Physics, 2019_

## Referee Comment (RC1) · Anonymous Referee #1 · 6 Feb 2019

This Hollaway et al. manuscript is tightly written and well put together. As stated in their Discussion (P12/L20), this new work "highlights the sensitivity of the responses in photolysis rates not only to the vertical distribution of aerosol but also the chemical speciation of the particulates"

The measurements of aerosol pollution in the north of Beijing combined with the careful modeling are able to demonstrate the accuracy of the modeling of photolysis rates and hence the ability to use a model to calculate the aerosol-pollution influence on near surface ozone. Their conclusion seem solid and important in understanding just how aerosol pollution interacts with oxidants. I recommend publication after the authors consider the minor suggestions below.

The impact of this work would be greater if some editorial aspects are cleaned up.

[Figure]

P1/Abstract. I urge you to drop the decimal place on these per cents. IFor example, think that 40-60% adequately describes 40.4-66.2%. Also, I think that the key statement quoted above (P12/L20ff) should appear in the abstract. This is a powerful result and should be up front.

P2/L5 I would have expected the original 2 papers that calculated the aerosol photolytic effects to be noted here: Martin et al, and Bian et al., both 2003.

P3/L12 Curious. Why in 'North' capitalized here? in US English, I would think not.

P4/L1 I read through the Whalley 2018 paper and looked up their supplemental data and cannot find any source of cross section data for photolysis. Is this the correct source?

P4/L13 does non-refractory aerosol include semi or partly volatile SOA?

P4/L31 Eqn I am trying to understand the units here. b (extinction) should be in 1/Mm, right? But [X] is usually a concentration unit (#/cm3). So please assign units carefully to all to help this reader.

P5/L3-12 Very nice design. I like the careful merger to get consistent measurement data for the modeling.

P5/L19 'account for' does not make sense to me, do you mean "average over" ? Can you specify what options/versions you used in Cloud-J, IF you implemented observed cloud fraction? Cloud fraction is not mentioned here, so make it clear that you just used a single column atmosphere, full cloud or clear in each layer.

P5/L21-33 This is a nice augment to the Fast-J code, boththe NO2 absorption and the aerosol cross sections. Are they available as a mod to a Fast-J/Cloud-J version? They ought to be. Using H-G and asymmetry parameter is OK for generating the phase function, but Mie would be better (outside the scope of this paper I know).

P6/L8-27 The high-frequency comparison in Figure 1 is fascinating and it is interesting

to see the mean bias over the diurnal cycle. Since you are using the same cross sections for both, it means that FJX is too hot, and overestimates the high-sun fluxes. Can you check if this holds for clear, unpolluted days?

Also, I think it would be valuable here to add something like a correlation coefficient to test if FJX+'Observed aerosols' can match the daily variability. Just add the r2 to the right-hand figure.

Use of %: Here is where the decimal point in the % numbers makes some sense (i.e., the bias). There are so many % numbers in this paper, it would be good to try to differentiate them simply. Otherwise the unit '%' should be fleshed out to say '% of what'. The % in a bias should maybe always have a sign: e.g., L13, +5.6 % mean bias above observations. See next section.

P6/L32 Use of %: Maybe do not need % here, but always need units to be clear. "In winter (NH4)2SO4 (39% of all aerosols by mass? by number? by optical depth?) and BC (30%) provide...

P7/L2 & Fig 2 "Vertical AOD profiles" and Fig 2 (bottom) make no sense in terms of units and what is plotted. The top row of Fig 2, it is the fraction of AOD from each component. Here the extra decimal point makes sense and does not clutter the reading. The problem is that AOD is always an extrinsic quantity while extinction (b, beta) is intrinsic. AOD is always integrated over a column or path length, but b is local. Thus you can plot b vs altitude, but not AOD. Please fix units in plot. See also L17 with "AOD values", and L20.

P7/L9 Use of %: here and elsewhere in this discussion, please round off. 20% is just fine instead of 19.8%, since the uncertainty is certainly greater than +-1%.

P7/L27 Use of % again: We have 23.8 and 23.1% reductions, the extra decimal is not meaningful in this discussion. Also the units are not clear. J's are about -23 % below what? a clear sky? a sky with all the other pollution but only those removed? If the
latter, it does raise the question of linearity or interference across the aerosol mix.

All throughout this discussion the nearest % is more than adequate. P8L16 & Fig 4. I wonder if this figure should show absolute changes in J's instead of %. I would think that this would emphasize the reactivity better, since % changes at low sun are not really important. For something like J-HONO, this would be fine, since the J's are more of a square wave. Figures 3 & 5 are fine as is, even if you change this to absolute deltas. If you want, you could do % of some noontime mean.

P8L30-35 Yes, this is an important result. Aerosol-pollution scattering and absorption above the boundary layer could be the most important factor.

P9/L5+ You can note here that – as you have found for polluted boundary layer - clouds have a much greater relative impact on J[NO2] vs J[O1D] also in observations over the clean remote Pacific [Hall et al., acp-18-16809-2018, very new paper, also using FJX, not available at time of drafting this paper.]

P9/L11-17 I found this paragraph confusing, and could not get the message.

P9-P12 Discussion This section is long and to me it wanders. If this discussion is useful, please do some numbered sub-sectioning for the reader.

P10/L27 "would be balanced by . . . rise ion NO. . ." To me this is not logical, since more NO means that less of O3 is tied up as NO2 in the NOx emissions, and further, more NO would enhance the ROO+NO reactions? Does this not augment the enhanced production rates? as oppose to balance them?

P11/L6 Use of %: "around 12.0%", really. "by 12 % and 3 %, respectively."

P13/L10 It seems that I have read something like this before. Do you need to repeat?

P13/L27 I think you need to have both observational data, plus the FJX code (that part that was adapted to NO2) and aerosol scattering data for FJX. I hope you get a doi eventually because otherwise it would be impossible to find on the CEDA site.

---

## Referee Comment (RC2) · Anonymous Referee #2 · 14 Mar 2019

This manuscript assesses the impact of aerosol vertical distribution and composition on photolysis rates over Beijing using ground-based aerosol measurements from the Air Pollution and Human Health campaign and the offline Fast-JX photolysis code. The key result is that despite significant differences in aerosol composition between winter and summer, aerosols in Beijing tend to depress photolysis rates near the Earth's surface and enhance them above, with important implications for photochemical responses to PM2.5 mitigation strategies. The study is well-conceived and the manuscript is generally well-written. I recommend publication in Atmospheric Chemistry and Physics with minor revisions as detailed below.

Specific Comments:

Abstract: The abstract contains a lot of specific results, but some of the key outcomes

of the paper get a bit lost in all of the numbers. In addition, the implications receive short shrift. There are many implications regarding potential PM2.5 mitigation strategies that are discussed in Sections 4 and 5 but are not reflected in the abstract.

Page 2, line 1: Please be specific about what is meant by "very high levels"

Page 2, line 29-30: the ")" is missing at the end of "(e.g. strong absorbers such as BC….."

Page 3, lines 25-26: How were the extinction coefficients attributed specifically to anthropogenic aerosols?

Page 5, lines 23-24: Is "cloud cover" equivalent to "cloud fraction"? If so, the latter term might be clearer since that is (I believe) what is used in Fast-JX.

Page 6, Section 3.1: There is one instance in the entire record where Fast-JX fails to model a significant decrease in photolysis rates when one is observed – May 29th. Are the authors able to comment on what was special about this particular day?

Page 6, line 29: Please add "of each aerosol component" after "vertical profiles and contributions"

Page 7, lines 1-2: The wording here makes it seem as though NH4NO3 and organic aerosol make similar contributions as (NH4)2SO4.

Page 7, lines 2-4: I'm not sure I agree with the authors' characterization of the vertical distribution. There is an enhancement of aerosol from 1-2 km that is not as large as in the boundary layer and above 3 km, but does not seem to be consistent with "high values … below 1 km which then decline rapidly with altitude before peaking again above 3 km"

Page 7, lines 14-16: Can the authors comment on why organic aerosol does not show the same vertical profile as the other aerosols? While the peak values are indeed within the same altitude range as the EPL, there is no layer-like feature in the OA.

Page 7, line 31: The word "substantial" has a typo

Page 7, lines 31-33: Why is the impact on J[NO2] larger than that on J[O1D]?

Page 7, line 33 – Page 8, line 2: The authors state that the surface layer is below the elevated levels of aerosol, but (NH4)2SO4 and BC are still clearly elevated at the surface according to the left panel of Figure 2. Please clarify.

Page 8, lines 6-7: Here the results in Figure 3 are attributed to "high levels of backscatter from the EPL", but again the OA in Figure 2, 3rd panel does not seem to show a distinct layered structure like the other aerosols do. It seems more accurate to say "from the level of maximum OA" or something similar.

Page 8, lines 19-21: Can the authors elaborate on why the effects of scattering are less pronounced for J[O1D]?

Page 8, lines 31-34: I found the statement that "particulate matter confined mainly to the boundary layer is shown to produce significant impacts at altitude" to be confusing - in much of the preceding discussion, many of the features of Figure 3 were attributed to the EPL during each season, and during summer the EPL seems to dominate the photolysis rate response. I do not see in the analysis provided any example where there is aerosol confined mainly to the boundary layer on which to base this statement.

Page 9, line 18: It would be helpful to the reader to include a "(not shown)" in the first sentence of the paragraph.

Page 10, line 20: Please explain briefly what is meant by 'photochemical limitation'.

Page 10, lines 24-30: It is clear that China has, in fact, implemented some emissions controls on aerosol precursors (see, for example, Wang et al., ERL, 2015, doi: 10.1088/1748-9326/10/11/114015 and Liu et al., ERL, 2016, doi:10.1088/1748-9326/11/11/114002) and satellite measurements show rapid decreases in NO2 and SO2 from 2011 onward. China's clean air plans (including specific targets for Beijing) should at least be mentioned here. In addition, one of the key points of this study is

that the aerosol composition matters, but that aspect is missing here. The differences in aerosol composition between summer and winter appear to make it very unlikely that a single mitigation approach would have substantial impacts in both seasons. Yet the scenarios described here simply assume "reduction of aerosol composition" without regard to species.

Page 11, lines 5-14: Please see the previous comment regarding the lack of a discussion of aerosol composition. Given the focus on composition in this paper, it seems odd to treat aerosol as a singular component.

Page 11, lines 25-26: The wording here ("with contrasting results") is very unclear.

Pages 11-13, Sections 4 and 5: There is some repetition between sections here that could be reduced.

Page 13, lines 15-16: I think it would be clearer and more impactful to explicitly say that reducing aerosols (which has a health benefit) would lead to more ozone at the surface (which has a negative health impact) – rather than simply "would offset the photochemical impacts demonstrated here".
* * *

---

## Author Comment (AC1) · 14 May 2019

**Responses to reviewer comments on 'Photochemical impacts of haze pollution in an Urban Environment'.**

The authors would like to thank both reviewers for their detailed and useful comments on our manuscript. Please find below our responses to each reviewers comment. The comment from the reviewer is shown in italic and our response is shown in bold. We also highlight changes in the revised manuscript in red.

**Anonymous reviewer 1:**

*P1/Abstract. I urge you to drop the decimal place on these per cents. I For example, think that 40-60% adequately describes 40.4-66.2%. Also, I think that the key statement quoted above (P12/L20ff) should appear in the abstract. This is a powerful result and should be up front.*

**We thank the reviewer for this suggestion. We have removed the decimal place from the abstract for clarity, and have also amended this in other places in the paper. We have also added the following key statement to the end of the abstract:**

**'Idealised photochemical box model studies show that such large impacts on photochemistry could lead to a 12% reduction in surface $O_3$ (3% for OH) due to haze pollution. This therefore highlights that any $PM_{2.5}$ mitigation strategies could have important implications for the oxidation capacity of the atmosphere both at the surface and in the free troposphere.'**

*P2/L5 I would have expected the original 2 papers that calculated the aerosol photolytic effects to be noted here: Martin et al, and Bian et al., both 2003.*

**We agree that this was an oversight and have added the suggested references on the photolytic effects of aerosols.**

*P3/L12 Curious. Why in 'North' capitalized here? in US English, I would think not.*

**We thank the reviewer for spotting this error. North should not be capitalised here and has been corrected in the revised manuscript.**

*P4/L1 I read through the Whalley 2018 paper and looked up their supplemental data and cannot find any source of cross section data for photolysis. Is this the correct source?*

**This method used to determine measured photolysis rates in the Beijing campaigns was the same as used in the ClearFlo project described by Whalley et al 2018 hence why we have cited this paper here. The sources of the cross sections and quantum yields are IUPAC and JPL.**

*P4/L13 does non-refractory aerosol include semi or partly volatile SOA?*

**The Aerosol mass spectroscopy (AMS) data only provided speciation of aerosol to sulphate, nitrate chloride and lumped organic aerosol. The lumped organic component will contain semi volatile SOA. This has been explored through PMF analysis in other parts of the APHH Beijing Programme however for the purposes of this study we have used the lumped OA in the Fast-JX simulations.**

*P4/L31 Eqn I am trying to understand the units here. b (extinction) should be in 1/Mm, right? But [X] is usually a concentration unit (#/cm3). So please assign units carefully to all to help this reader.*

We apologise for the confusion here. As described in the original paper defining the IMPROVE algorithm for attributing extinction to aerosol species (Pitchford et al., 2007) the coefficients in equation 1 are the mass scattering/absorption efficiency (MSE/MAE) for each aerosol species. These have units of $m^2/g$. In our case our aerosol are specified as mass concentrations ($\mu g/m^3$) and therefore will return units of inverse megametres ($Mm^{-1}$) for the extinction. We have clarified this in the revised manuscript.

*P5/L3-12 Very nice design. I like the careful merger to get consistent measurement data for the modeling.*

**Thanks very much.**

*P5/L19 'account for' does not make sense to me, do you mean "average over" ?*

**This sentence was intended to indicate that in order to account for multiple layers of overlapping cloud in the atmospheric column we use a quadrature approach to average over cloud fractions in all layers. On the reviewers' suggestion we have changed 'account for' to 'average over' in the revised manuscript.**

*Can you specify what options/versions you used in Cloud-J, IF you implemented observed cloud fraction? Cloud fraction is not mentioned here, so make it clear that you just used a single column atmosphere, full cloud or clear in each layer.*

**We implemented the quadrature approach (Neu et al 2007, Prather et al 2015) for cloud cover here using to account for overlapping cloud layers in the column. The cloud cover fraction product from the ERA5 (with cloud fractions specified for each model layer) reanalysis product was used to implement cloud cover in FJX. We have clarified this in the revised manuscript.**

*P5/L21-33 This is a nice augment to the Fast-J code, both the NO2 absorption and the aerosol cross sections. Are they available as a mod to a Fast-J/Cloud-J version? They ought to be. Using H-G and asymmetry parameter is OK for generating the phase function, but Mie would be better (outside the scope of this paper I know).*

**The modifications made for the NO₂ absorption are not currently available as a mod to Fast-J/Cloud-J but we agree that they could be implemented into future versions of the code. The Mie approach would provide a more accurate method for generating the phase function but we have chosen a simpler approach for these studies at the present time due to the Mie approach being more computationally intensive.**

*P6/L8-27 The high-frequency comparison in Figure 1 is fascinating and it is interesting to see the mean bias over the diurnal cycle. Since you are using the same cross sections for both, it means that FJX is too hot, and overestimates the high-sun fluxes. Can you check if this holds for clear, unpolluted days?*

***FJX does appear to have a positive bias on both polluted and unpolluted days as shown in Figure 1. However, the availability of observed column ozone data was limited during both campaigns and therefore we used column ozone from the ERA5 reanalysis product. It is likely that the positive bias in FJX is related to biases in the ERA5 ozone column.***

*Also, I think it would be valuable here to add something like a correlation coefficient to test if FJX+'Observed aerosols' can match the daily variability. Just add the r2 to the right-hand figure.*

**We thank the reviewer for their suggestion. We have added correlation coefficients for both campaigns to the right hand figures in both plots in Figure 1 in the revised manuscript.**

*Use of %: Here is where the decimal point in the % numbers makes some sense (i.e., the bias). There are so many % numbers in this paper, it would be good to try to differentiate them simply. Otherwise the unit '%' should be fleshed out to say '% of what'. The % in a bias should maybe always have a sign: e.g., L13, +5.6 % mean bias above observations. See next section.*

**We thank the reviewer for the suggestion. This has been corrected in the revised manuscript.**

*P6/L32 Use of %: Maybe do not need % here, but always need units to be clear. "In winter (NH4)2SO4 (39% of all aerosols by mass? by number? by optical depth?) and BC (30%) provide: : :*

**We have corrected this in the revised manuscript and have clarified that the % values are the contribution to total column optical depth.**

*P7/L2 & Fig 2 "Vertical AOD profiles" and Fig 2 (bottom) make no sense in terms of units and what is plotted. The top row of Fig 2, it is the fraction of AOD from each component. Here the extra decimal point makes sense and does not clutter the reading. The problem is that AOD is always an extrinsic quantity while extinction (b, beta) is intrinsic. AOD is always integrated over a column or path length, but b is local. Thus you can plot b vs altitude, but not AOD. Please fix units in plot. See also L17 with "AOD values", and L20.*

**We apologise for the potential confusion in this plot. The top row in the figure is the fraction of total column aerosol apportioned to each species. For improved clarity we have replaced the pie charts with bar charts and included the numbers in the main manuscript text. The lower plots show the integrated AOD over each 30m layer in the lidar profile however can appreciate the confusion with the units in the plot. For clarity we have replaced the lower plots with those that show extinction ($b_{ext}$) vs altitude. The units of extinction will be inverse megametres ($Mm^{-1}$). We have edited the text in the revised manuscript to reflect this updated figure.**

*P7/L9 Use of %: here and elsewhere in this discussion, please round off. 20% is just fine instead of 19.8%, since the uncertainty is certainly greater than +-1%.*

**We have corrected this here and throughout the discussion. All % changes have been rounded to the nearest whole number.**

*P7/L27 Use of % again: We have 23.8 and 23.1% reductions, the extra decimal is not meaningful in this discussion. Also the units are not clear. J's are about -23 % below what? a clear sky? a sky with all the other pollution but only those removed? If the latter, it does raise the question of linearity or interference across the aerosol mix. All throughout this discussion the nearest % is more than adequate.*

**All % changes have been rounded to the nearest % in this section. The impacts of each individual aerosol species have been calculated by including each species in isolation and comparing the changes in J rates to those simulated under clear sky conditions. Therefore the changes presented are the simulated effects from that particular species only. In the revised manuscript we clarify this by editing the first sentence of section 3.3 to read:**

**'As Fast-JX is run in offline mode, the effects of each aerosol species can be determined independently. Each aerosol species is allowed to influence incoming solar radiation in isolation (Table 1) allowing the change in photolysis rates with respect to clear sky conditions to be quantified (Figure 3). This allows quantification of the impacts of each species on photolysis rates during haze episodes in Beijing.'**

*P8L16 & Fig 4. I wonder if this figure should show absolute changes in J's instead of %. I would think that this would emphasize the reactivity better, since % changes at low sun are not really important. For something like J-HONO, this would be fine, since the J's are more of a square wave. Figures 3 & 5 are fine as is, even if you change this to absolute deltas. If you want, you could do % of some noontime mean.*

**We agree that % changes at low sun are of relatively low importance. We will change this figure to reflect absolute changes in J rates for both O₃ and NO₂ which will highlight the larger impacts during the middle of the data when photochemical activity is at its greatest.**

*P8L30-35 Yes, this is an important result. Aerosol-pollution scattering and absorption above the boundary layer could be the most important factor.*

**Thanks very much.**

*P9/L5+ You can note here that – as you have found for polluted boundary layer – clouds have a much greater relative impact on J[NO2] vs J[O1D] also in observations over the clean remote Pacific [Hall et al., acp-18-16809-2018, very new paper, also using FJX, not available at time of drafting this paper.]*

**We agree that our findings are consistent with that of Hall et al and have added the suggested reference to the revised manuscript.**

*P9/L11-17 I found this paragraph confusing, and could not get the message.*

**This paragraph aims to highlight that in haze events during the summer campaign clouds had a dominant effect on J-rates at the surface but that aerosols became increasingly important higher up the column. We have clarified this as follows in the revised manuscript.**

**"In summer, during haze conditions, clouds produce the largest impacts on photolysis rates in the surface layer (reductions of 10–11%), approximately double that attributed to aerosol (~6%). The effects of high levels of scattering aerosol (particularly OA) during these conditions are evident higher in the column where increases in photolysis rates due to aerosol are much larger than those from clouds. The combined effects of cloud and aerosol are reductions in the lowest 3 km (0.1–17.2% for J[O$^1$D] and 1.2–15.7% J[NO$_2$]) which are dominated by the influence of clouds. Between 3-6 km the effects of backscatter from aerosol are greater than those from clouds giving net increases in photolysis rates (8.8–13.7% for J[O$^1$D] and 11–18% J[NO$_2$])"**

*P9-P12 Discussion This section is long and to me it wanders. If this discussion is useful, please do some numbered sub-sectioning for the reader.*

**We thank the reviewer for this suggestion. We have edited the discussion slightly to tighten up the flow and readability of the discussion. This has been done by separating out the discussion on the simple box model studies into its own sub-section at the end of the discussion section in the revised manuscript.**

*P10/L27 "would be balanced by : : : rise ion NO: : :" To me this is not logical, since more NO means that less of O3 is tied up as NO2 in the NOx emissions, and further, more NO would enhance the ROO+NO reactions? Does this not augment the enhanced production rates? as oppose to balance them?*

**The enhanced $NO_2$ photolysis would produce more NO which would subsequently react with $O_3$ to form $NO_2$. However the reviewer is correct in pointing out that added NO would also enhance ROO+NO reactions which would lead to higher $O_3$ concentrations. We have clarified this point in the revised manuscript to confirm the net increase in $O_3$ from this pathway.**

**'Enhanced near-surface photolysis rates would also increase $O_3$ production via $NO_2$ photolysis and enhanced levels of NO. This rise in $O_3$ will be partially balanced by the reaction of NO with $O_3$ itself. However, higher NO levels will also contribute to enhanced $O_3$ formation through increases in $RO_2$ and NO reactions. Furthermore, the enhancement of $J[O^1D]$ would increase OH concentrations which would subsequently increase $HO_2$ and $RO_2$ and lead to a net rise in $O_3$ concentrations.'**

*P11/L6 Use of %: "around 12.0%", really. "by 12 % and 3 %, respectively."*

**We have corrected this in the revised manuscript.**

*P13/L10 It seems that I have read something like this before. Do you need to repeat?*

**This sentence was to highlight the improvement this study makes on previous all modelling based studies as part of the paper conclusions. However we appreciate it is repetitive and have therefore tightened up this paragraph to read as follows.**

**"The observation-driven approach to deriving the aerosol vertical distribution allows a more accurate constraint to be made on the estimated impacts of haze pollution on photochemistry and, more critically, allows species specific impacts to be highlighted. This allows the potential identification of source sectors to target particulate control strategies on. "**

*P13/L27 I think you need to have both observational data, plus the FJX code (that part that was adapted to NO2) and aerosol scattering data for FJX. I hope you get a doi eventually because otherwise it would be impossible to find on the CEDA site.*

**We will upload the model simulation data to CEDA when preparing the revised manuscript and will ensure that the doi is provided in the revised manuscript. Links will also be provided to the SP2 data (CEDA), aerosol AMS data (available from IAP on request) and aerosol extinction data (available from IAP on request). Links to the ERA5 data set will also be provided here.**

**Anonymous reviewer 2:**

*Abstract: The abstract contains a lot of specific results, but some of the key outcomes of the paper get a bit lost in all of the numbers. In addition, the implications receive short shrift. There are many implications regarding potential PM2.5 mitigation strategies that are discussed in Sections 4 and 5 but are not reflected in the abstract.*

**The main goal of this paper is to highlight the impacts of aerosols on photolysis rates during haze episodes rather than to investigate the potential impacts of PM$_{2.5}$ mitigation strategies. Therefore we have highlighted these key impacts in the abstract. We have edited the last sentence in the abstract highlighting the potential implications that removing aerosol could have on atmospheric oxidants.**

**'Idealised photochemical box model studies show that such large impacts on photochemistry could lead to a 12% reduction in surface O$_3$ (3% for OH) due to haze pollution. This therefore highlights that any PM$_{2.5}$ mitigation strategies could have important implications for the oxidation capacity of the atmosphere both at the surface and in the free troposphere.'**

*Page 2, line 1: Please be specific about what is meant by "very high levels"*

**In this case we are referring to particulate matter concentrations of greater than 75 μgm$^{-3}$ which corresponds to an AQI of 100. We have clarified this in the revised manuscript.**

*Page 2, line 29-30: the ")" is missing at the end of "(e.g. strong absorbers such as BC: : :.."*

**We have corrected this in the revised manuscript**

*Page 3, lines 25-26: How were the extinction coefficients attributed specifically to anthropogenic aerosols?*

**The lidar is dual wavelength and measures depolarisation, and attribution of measured backscatter to anthropogenic aerosol was made using the depolarisation ratio. This is described in Yang et al (2010, 2017) who describe the instrument in detail. To clarify this we have altered the final sentence in this paragraph as follows:**

**'Further details of the lidar instrument, calibration procedures and attribution of extinction coefficients to anthropogenic aerosol using the depolarisation ratio can be found in Yang et al (2010, 2017) and Sugimoto et al. 2002.'**

*Page 5, lines 23-24: Is "cloud cover" equivalent to "cloud fraction"? If so, the latter term might be clearer since that is (I believe) what is used in Fast-JX.*

**We have used cloud cover here to refer to the fraction of each layer in the column that is covered by cloud. To clarify this we have revised the manuscript to use cloud fraction.**

*Page 6, Section 3.1: There is one instance in the entire record where Fast-JX fails to model a significant decrease in photolysis rates when one is observed – May 29th. Are the authors able to comment on what was special about this particular day?*

**As the model is being driven using ERA5 reanalysis cloud fraction data, this failure to capture the drop in photolysis rates on the 29$^{th}$ May is likely to be due to misrepresentation of the cloud cover in this dataset. Therefore, as Fast-JX is being driven in offline mode, misrepresentation in the**

**cloud fields will lead to errors in the modelled J rates and thus the model's failure to capture the decrease seen in the observations.**

*Page 6, line 29: Please add "of each aerosol component" after "vertical profiles and contributions"*

**We have corrected this in the revised manuscript. Please also see our response to reviewer 1 where we have edited the vertical profiles in the figure to show extinction coefficient rather than AOD. This sentence now reads:**

'**Figure 2 shows vertical profiles of extinction coefficient for each aerosol component for both campaigns as derived from the optimisation approach described in Section 2.3. Figure 2 also shows the contribution to total column AOD of each aerosol component.**'

*Page 7, lines 1-2: The wording here makes it seem as though NH4NO3 and organic aerosol make similar contributions as (NH4)2SO4.*

**The intention here was to indicate that $NH_4NO_3$ and organic aerosol make similar contributions to each other but not quite as large as those as for $(NH_4)_2SO_4$ and BC. We have rephrased this in the revised manuscript.**

*Page 7, lines 2-4: I'm not sure I agree with the authors' characterization of the vertical distribution. There is an enhancement of aerosol from 1-2 km that is not as large as in the boundary layer and above 3 km, but does not seem to be consistent with "high values : : : below 1 km which then decline rapidly with altitude before peaking again above 3 km"*

**In the revised manuscript we have clarified the description of the vertical distribution to indicate that aerosol increases slightly between 1-2 km but this is not as high as values seen in the BL or above 3 km. This sentence reads as follows in the revised manuscript.**

"**Vertical aerosol extinction profiles show large peaks in both the boundary layer (below 1 km) and above 3 km where there is evidence of an elevated pollution layer (EPL). Elsewhere in the column extinction values are much lower however there are slightly elevated values at 1-2 km, although these are not as large as seen in the boundary layer or EPL.**"

*Page 7, lines 14-16: Can the authors comment on why organic aerosol does not show the same vertical profile as the other aerosols? While the peak values are indeed within the same altitude range as the EPL, there is no layer-like feature in the OA.*

**The OA does indeed exhibit a different vertical profile to the other species. There are high values in the elevated layer as for other species, but the structure is different, and this suggests that the sources may be rather different (e.g. the influence of biogenic species on the formation of secondary organic aerosol). It could also be linked to factors in the optimisation algorithm which determines the mean scattering efficiency (MSE) of the organic aerosol (Equation 1) and attributes the lidar extinction to each aerosol species.**

*Page 7, line 31: The word "substantial" has a typo*

**We have corrected this in the revised manuscript.**

*Page 7, lines 31-33: Why is the impact on J[NO2] larger than that on J[O1D]?*

**Generally NO₂ photolysis is more sensitive to scattered radiation than that of O₃ and other species in particular with increasing height above layers of scattering aerosol. This is likely due to the longer wavelength dependence of JNO₂. As JNO₂ occurs largely in the visible part of the spectrum and path length increases with height from the scattering layer, the effects on NO₂ will be amplified compared to those of J[O¹D]. This effect is also present at sunrise and sunset.**

*Page 7, line 33 – Page 8, line 2: The authors state that the surface layer is below the elevated levels of aerosol, but (NH4)2SO4 and BC are still clearly elevated at the surface according to the left panel of Figure 2. Please clarify.*

**We apologise for the confusion here. The reviewer is correct that aerosol concentrations remain elevated at the surface. This sentence was intended to reflect this and has been corrected in the revised manuscript to read:**

**"In the surface layer, within the elevated levels of aerosol, the scattering aerosol lead to reductions of -1.7% to -4.4% for J[O¹D] and -3.4% to -7.0% for J[NO₂], with OA producing the largest reduction in both cases."**

*Page 8, lines 6-7: Here the results in Figure 3 are attributed to "high levels of backscatter from the EPL", but again the OA in Figure 2, 3rd panel does not seem to show a distinct layered structure like the other aerosols do. It seems more accurate to say "from the level of maximum OA" or something similar.*

**This is correct, the dominant response is from the high maximum level of OA at around 4 km (Figure 2) which is also where the other aerosols exhibit their maximum extinctions in the summer. We have now clarified this sentence to read as follows in the revised manuscript.**

**"This represents the high levels of backscatter from the level of maximum OA which occurs at a height of approximately 4 km (Figure 2)."**

*Page 8, lines 19-21: Can the authors elaborate on why the effects of scattering are less pronounced for J[O1D]?*

**This is due to the longer wavelength dependency of JNO₂ compared to that of J[O¹D]. Therefore with increasing path length from the scattering layer the relative impacts on JNO₂ are amplified.**

*Page 8, lines 31-34: I found the statement that "particulate matter confined mainly to the boundary layer is shown to produce significant impacts at altitude" to be confusing - in much of the preceding discussion, many of the features of Figure 3 were attributed to the EPL during each season, and during summer the EPL seems to dominate the photolysis rate response. I do not see in the analysis provided any example where there is aerosol confined mainly to the boundary layer on which to base this statement.*

**This statement highlights that the majority of aerosol sources are within the boundary layer. Although most of the impacts of haze pollution (reductions in visibility, health impacts, etc) are at the surface, there is a significant non-local effect of these aerosols at altitude through changes in photolysis rates. We have clarified this sentence in the revised manuscript to reflect this.**

*Page 9, line 18: It would be helpful to the reader to include a "(not shown)" in the first sentence of the paragraph.*

This has been corrected in the revised manuscript.

*Page 10, line 20: Please explain briefly what is meant by 'photochemical limitation'.*

The term 'photochemical limitation' is used here to indicate that the rate of ozone formation is largely dependent on the amount of incoming radiation. Therefore in the summer months higher incident solar radiation results in higher levels of ozone formation. For clarity, we have amended this sentence in the revised manuscript to read as follows.

**"This is important as incoming solar radiation is at its highest during the summer months which results in higher rates of $O_3$ formation (Tie and Cao, 2009)."**

*Page 10, lines 24-30: It is clear that China has, in fact, implemented some emissions controls on aerosol precursors (see, for example, Wang et al., ERL, 2015, doi: 10.1088/1748-9326/10/11/114015 and Liu et al., ERL, 2016, doi:10.1088/1748-9326/11/11/114002) and satellite measurements show rapid decreases in NO2 and SO2 from 2011 onward. China's clean air plans (including specific targets for Beijing) should at least be mentioned here. In addition, one of the key points of this study is that the aerosol composition matters, but that aspect is missing here. The differences in aerosol composition between summer and winter appear to make it very unlikely that a single mitigation approach would have substantial impacts in both seasons. Yet the scenarios described here simply assume "reduction of aerosol composition" without regard to species.*

The reviewer is correct that China's clean air plans, have seen reduction in aerosol concentrations, in particular the recent drop in $PM_{2.5}$ that has been seen over Beijing. These reductions in particulate concentrations will reduce the attenuation of radiation which in turn will enhance $J[O^1D]$ and $JNO_2$ at the surface. This will enhance $O_3$ formation through higher NO levels from $JNO_2$ photolysis which enhance ozone concentrations through ROO+NO reactions and NO reaction with $O_2$. Furthermore, enhanced $J[O^1D]$ will lead to higher OH concentrations which can lead to $NO_x/HO_x$ cycling resulting in $O_3$ formation. We have therefore edited this paragraph to read as follows in the revised manuscript.

**'Therefore, under the recent ~30% reduction in $PM_{2.5}$ in Beijing through emissions controls implemented as part of China's clean air plans (Wang et al., 2015; Liu et al., 2016), the resultant increases in $J[O^1D]$ could potentially lead to enhanced $O_3$ concentrations, where summer levels are already very high (Wang et al., 2006; Xue et al., 2014; Ni et al., 2018). Enhanced near-surface photolysis rates would also increase $O_3$ production via $NO_2$ photolysis though this will be partially balanced by the reaction of NO with $O_3$ itself. However, higher NO levels will also contribute to enhanced $O_3$ formation through increases in $RO_2$ and NO reactions. Furthermore, the enhancement of $J[O^1D]$ would increase OH concentrations which would subsequently increase $HO_2$ and $RO_2$ and lead to a net rise in $O_3$ concentrations. In the winter, a similar response would be expected, although due to lower photolysis rates and much lower $O_3$ concentrations, the effects of particulate control strategies would have a lesser effect on oxidant concentrations.'**

With respect the reviewers' comment regarding mitigation strategies targeting individual species, the main aim of this study is to look at the impacts of haze pollution on photolysis rates. The photochemical box model simulation are used as an indication of the potential chemical impacts the presence of aerosols in the urban atmosphere could have. As it is not within scope of the present study we are not evaluating the impacts of different mitigation strategies (and thus strategies targeting individual aerosol species) we do not discuss this here.

*Page 11, lines 5-14: Please see the previous comment regarding the lack of a discussion of aerosol composition. Given the focus on composition in this paper, it seems odd to treat aerosol as a singular component.*

**As the main focus of this study is to focus on the impacts of haze pollution on photolysis rates and not evaluating the impacts of different mitigation strategies we have not focussed on aerosol composition in the simple box model simulations. The main aim of this discussion is to focus on the potential overall effect of aerosols on photochemistry. Due to the complex non-linearities involved in the chemistry a full attribution effect to each aerosol species is beyond the scope of this study. Further to this any aerosol control strategy is likely to reduce all species rather than just one therefore the reduction of all aerosol in the simple box model study is appropriate in this case.**

*Page 11, lines 25-26: The wording here ("with contrasting results") is very unclear.*

**We appreciate that this is unclear and have removed the wording "contrasting results" from the sentence in the revised manuscript.**

*Pages 11-13, Sections 4 and 5: There is some repetition between sections here that could be reduced.*

**We have tightened up these sections in the revised manuscript to remove the repetition. We have also improved the flow and clarity of the discussion by separating out the box model results into its own sub-section.**

*Page 13, lines 15-16: I think it would be clearer and more impactful to explicitly say that reducing aerosols (which has a health benefit) would lead to more ozone at the surface (which has a negative health impact) – rather than simply "would offset the photochemical impacts demonstrated here".*

**Implementing particulate control strategies not only reduces concentrations of aerosols but also the corresponding impacts in photolysis rates. The box model results suggest that this will enhance ozone at the surface but due to the complex non-linearities involved a full quantification of the impacts on ozone concentrations is needed. We have clarified this sentence in the revised manuscript to read.**

**"Such strategies would not only reduce particulate matter concentrations, but also reduce their impacts on photolysis rates and thus potentially increase surface ozone concentrations."**

**For clarity we have also separated the conclusions section about the chemical box model results into a new paragraph.**

**References**

Liu, F., Zhang Q., van der A R. J., Zheng, B., Tong D., Yan L., Zheng Y., and He K., Recent reduction in NOx emissions over China: synthesis of satellite observations and emission inventories, Environmental Research Letters, 11, 114002, doi:10.1088/1748-9326/11/11/114002, 2016.

Ni, R., Lin, J., Yan, Y., and Lin,W.: Foreign and domestic contributions to springtime ozone over China, Atmospheric Chemistry and Physics, 18, 11 447–11 469, https://doi.org/10.5194/acp-18-11447-2018, 2018.

Sugimoto, N., Matsui, I., Shimizu, A., Uno, I., Asai, K., Endoh, T., and Nakajima, T.: Observation of dust and anthropogenic aerosol plumes in the Northwest Pacific with a two-wavelength polarization lidar on board the research vessel Mirai, Geophysical Research Letters, 29,7–1–7–4, https://doi.org/10.1029/2002GL015112, 2002.

Tie, X. and Cao, J.: Aerosol pollution in China: Present and future impact on environment, Particuology, 5 7, 426 – 431, https://doi.org/https://doi.org/10.1016/j.partic.2009.09.003, 2009.

Wang, T., Ding, A., Gao, J., and Wu, W. S.: Strong ozone production in urban plumes from Beijing, China, Geophysical Research Letters, 33, https://doi.org/10.1029/2006GL027689, 2006.

Wang, S., Zhang, Q., Martin, R.V., Philip, S., Liu, F., Li M., Jiang X., He, K., Satellite measurements oversee China's sulfur dioxide emission reductions from coal-fired power plants, Environmental Research Letters, 10, 114015, doi:10.1088/1748-9326/10/11/114015, 2015.

Xue, L. K., Wang, T., Gao, J., Ding, A. J., Zhou, X. H., Blake, D. R., Wang, X. F., Saunders, S. M., Fan, S. J., Zuo, H. C., Zhang, Q. Z., and Wang, W. X.: Ground-level ozone in four Chinese cities: precursors, regional transport and heterogeneous processes, Atmospheric Chemistry and Physics, 14, 13 175–13 188, https://doi.org/10.5194/acp-14-13175-2014, 2014.

Yang, T., Wang, Z., Zhang, B., Wang, X., Wang, W., Gbauidi, A., and Gong, Y.: Evaluation of the effect of air pollution control during the Beijing 2008 Olympic Games using Lidar data, Chinese Science Bulletin, 55, 1311–1316, https://doi.org/10.1007/s11434-010-0081-y, 2010.

Yang, T., Wang, Z., Zhang, W., Gbaguidi, A., Sugimoto, N., Wang, X., Matsui, I., and Sun, Y.: Technical note: Boundary layer height determination from lidar for improving air pollution episode modeling: development of new algorithm and evaluation, Atmospheric Chemistry and Physics, 17, 6215–6225, https://doi.org/10.5194/acp-17-6215-2017, 2017.